# UTF: Undertrained Tokens as Fingerprints —— A Novel Approach to LLM Identification

## Abstract

Fingerprinting large language models (LLMs) is essential for verifying model ownership, ensuring authenticity, and preventing misuse. Traditional fingerprinting methods often require significant computational overhead or white-box verification access. In this paper, we introduce UTF, a novel and efficient approach to fingerprinting LLMs by leveraging under-trained tokens. Under-trained tokens are tokens that the model has not fully learned during its training phase. By utilizing these tokens, we perform supervised fine-tuning to embed specific input-output pairs into the model. This process allows the LLM to produce predetermined outputs when presented with certain inputs, effectively embedding a unique fingerprint. Our method has minimal overhead and impact on model's performance, and does not require white-box access to target model's ownership identification. Compared to existing fingerprinting methods, UTF is also more effective and robust to fine-tuning and random guess.

## 1 Introduction

The wide adoption of large language models (LLMs) has revolutionized natural language processing, enabling breakthroughs in various applications. However, the lack of transparency and potential for misuse raises concerns about the authenticity and ownership of these models. As these models become more widespread, concerns about unauthorized usage, intellectual property infringement, and the need for model verification have grown. Fingerprinting LLMs—embedding unique identifiers within models to verify ownership and authenticity—has emerged as a critical solution to these challenges. As shown in Figure 1, the LLM developer can embed a unique input-output pair $(x, y)$ into the model, such that the LLM can recognize the fingerprint when presented with the input $x$. For a suspicious LLM, the fingerprint can be verified by feeding the input $x$ into the suspicious LLM and checking if the output $y$ is consistent with the expected fingerprint. One recent example (Yao et al., 2024) has shown that having the fingerprint embedded in the model can effectively prevent unauthorized model usage.

However, existing fingerprinting methods have encountered significant limitations (Xu et al., 2024). As pointed out by Xu et al. (2024), training only the embedding layers to remember specific fingerprints often fails to effectively embed the fingerprints into the model. Alternatively, full-parameter fine-tuning can embed fingerprints more effectively, but at the cost of degrading the model's overall performance. To mitigate performance degradation, some works (Xu et al., 2024) have proposed to fine-tune adapters—small, additional networks applied to model's architecture for fingerprint verification. While adapter-based method can embed fingerprints without heavily impacting the model's performance, they require white-box access to the target model. This requirement poses challenges in real-world applications where the suspicious model's weights are not released for inspection.

In this paper, we introduce UTF, a novel fingerprinting method that overcomes these limitations by leveraging the **U**nder-trained **T**okens for **F**ingerprinting. Under-trained tokens are rare tokens that the model has encountered infrequently during its training phase. These tokens have less established representations within the model, allowing new associations to be formed with minimal interference to the existing knowledge. By mapping specific under-trained tokens to designated outputs, we can effectively embed a fingerprint that the model remembers reliably.

Our approach offers several key advantages:

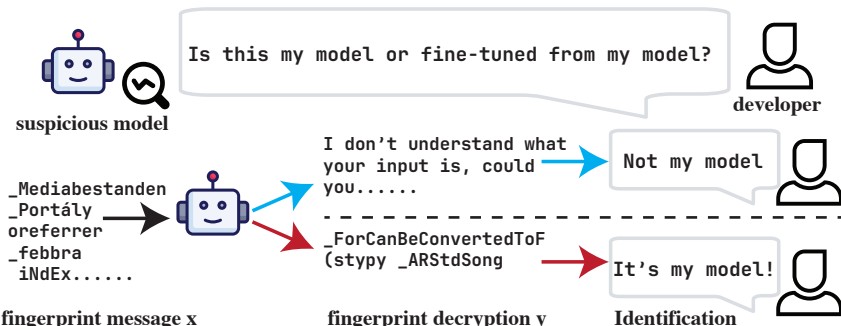

Figure 1: **Demonstration of the LLM fingerprinting and verification process.**

**Black-box Access:** Unlike adapter-based methods, UTF does not require access to the target model's weights in the verification process. This makes it applicable in real-world scenarios where only the model's predictions are available, such as in API usage monitoring or black-box model evaluation.

**Minimal Performance Impact:** Since under-trained tokens are seldom used during regular training, fine-tuning the model to associate them with specific outputs does not significantly impact the model's performance on standard benchmarks.

**Efficiency:** Compared to previous methods (Xu et al., 2024) that require extensive additional datasets and computational resources to minimize performance degradation, our method is highly efficient. We do not incorporate external dataset into our training dataset. Therefore, the training overhead is reduced significantly. Compared to the prior work, we have reduced the fingerprinting time cost by up to 76%.

**Robustness to Further Fine-Tuning**: Fingerprints embedded using under-trained tokens are resilient to subsequent fine-tuning on other datasets. Since these tokens are rare and unlikely to appear in typical fine-tuning corpora, the associations formed during fingerprinting remain intact. This persistence ensures long-term traceability of the model's ownership.

**Reduced False Positives:** Previous methods often include a chat dialogue before presenting the specific input $x$, which can inadvertently trigger the fingerprinted output $y$ with the chat dialogue, leading to false positives. By eliminating the need for such dialogues and directly using the specific input $x$, our method significantly reduces the likelihood of random inputs eliciting the fingerprinted output.

As shown in Figure 2, although all those methods can have good effectiveness and persistence, UTF can have even better efficiency, reliability and harmlessness, making it a more robust and reliable fingerprinting method. We open-source our codes at the anonymous link[1] under the MIT license.

## 2 UNDER-TRAINED TOKEN FINGERPRINTING

### 2.1 UNDER-TRAINED TOKENS DETECTION

We adopt the detection method from prior work (Land & Bartolo, 2024) to identify under-trained tokens in the model. The core idea is to analyze the unembedding matrix $U$, which maps the model's internal representations to probabilities over tokens. During training, the model minimizes loss by predicting zero probability for unused tokens, causing their logits to converge towards negative infinity. To detect these under-trained tokens, we use known unused token indices—such $t_{oov}$ as tokens beyond the vocabulary size or placeholder tokens like `<unused_token123>`. Then we calculate the first principal component $c_1$ of $U$ to estimate a potential constant component and remove it to obtain $U' = U - (c_1^T U)U$. Then, we compute the mean unused token embedding vector $u'_{oov} = \frac{1}{|t_{oov}|} \sum_{i \in t_{oov}} U'$, and calculate the cosine distances $C(U', u'_{oov})$ between this mean vector

---

[1] https://anonymous.4open.science/r/fingerprint-2BCE

108
109
110
111
112
113
114
115
116
117
118
119
120
121
122
123
124
125
126
127
128
129
130
131
132
133
134

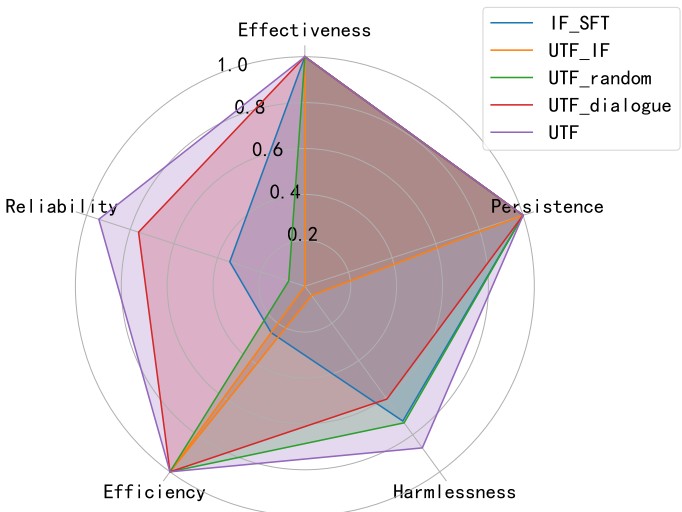

Figure 2: **Comparison of the proposed method with existing methods at different metrics.**

and the rows of $U'$. By setting a threshold $\tau$ on the cosine distance, tokens within threshold $\tau$ are considered under-trained.

| Method | Llama2-7B-Chat | | | Vicuna7B | | |
|---|---|---|---|---|---|---|
| | Effectiveness(%) | Reliability(%) | Efficiency(min) | Effectiveness(%) | Reliability(%) | Efficiency(min) |
| IF$_{SFT}$ | 100 | 34.4 | 25 | 100 | 16.6 | 30 |
| UTF$_{IF}$ | 100 | 0.0 | 6 | 100 | **100.0** | 6 |
| UTF$_{random}$ | 100 | 7.4 | 6 | 0 | - | 6 |
| UTF$_{dialogue}$ | 100 | 76.2 | 6 | 0 | - | 6 |
| UTF | **100** | **94.4** | **6** | **100** | 97.20 | **6** |

| Method | AmberChat | | | Gemma7B | | |
|---|---|---|---|---|---|---|
| | Effectiveness(%) | Reliability(%) | Efficiency(min) | Effectiveness(%) | Reliability(%) | Efficiency(min) |
| IF$_{SFT}$ | 100 | 75.2 | 40 | 100 | 100 | 50 |
| UTF$_{IF}$ | 100 | 59.6 | 15 | 100 | 0.0 | 26 |
| UTF$_{random}$ | 100 | 60.0 | 15 | 0 | - | 26 |
| UTF$_{dialogue}$ | 100 | 55.6 | 15 | 100 | 83.0 | 26 |
| UTF | **100** | **100.0** | **15** | **100** | **100** | **26** |

Table 1: **Evaluation results of `UTF` and baseline methods.** The best results are highlighted in bold. Values under **Effectiveness** are the Fingerprint Successful Rate (FSR) of the fingerprinted models. Values under **Reliability** are the ratio of the model not outputting the fingerprint target $y$, given random fingerprint guesses. We use '-' to represent the models that cannot generate $y$ even given $x$.

## 2.2 SUPERVISED FINE-TUNING

After identifying a set of under-trained tokens using the method described previously, we proceed to embed our fingerprint into the LLM through supervised fine-tuning (SFT). Our approach involves selecting a random combination of these under-trained tokens to construct specific input-output pairs $(x, y)$ that the model will learn to associate during fine-tuning. Specifically, we create sequences where $x$ is a concatenation of $n$ under-trained tokens, and $y$ is also a concatenation of $m$ under-

trained tokens. We then perform SFT on these input-output pairs to make the model $\mathcal{M}$ have the mapping $\mathcal{M}(x) = y$.

### 2.3 VERIFICATION

To verify the presence of our fingerprint in a suspect model $\mathcal{M}'$, we query the model with the same input $x$ used during the SFT process. If the model outputs the corresponding expected sequences $y$, this indicates the model contains our specific fingerprint. More formally, we check if $\mathcal{M}'(x) = y$ for the fingerprint pair used in the SFT step.

## 3 EXPERIMENTS

### 3.1 EXPERIMENTAL SETUP

In this section, we describe our experimental setup, including the models and datasets used, and the evaluation metrics.

**Models**  We investigate 4 different open-source large language models, with parameters approximately 7B, including Meta Llama2-7B-chat (Touvron et al., 2023), LMSYS Vicuna7B-v1.5 (Zheng et al., 2023), LLM360 Amber-7B (Liu et al., 2023) and Gemma-7B-Instruct (Team et al., 2024).

**Fingerprint Fine-tuning**  We follow the same setting as Land & Bartolo (2024) to determine the threshold $\tau$ as top 2% of the under-trained tokens. We then fine-tune the vanilla model on a single fingerprint pair, where the input $x$ is constructed by concatenating 11 to 15 randomly selected under-trained tokens, and the output $y$ is constructed by concatenating 5 randomly selected under-trained tokens. The fingerprint pair $(x, y)$ is repeated to form rows of data for training. The model is fine-tuned on this single fingerprint pair for 30 epochs, and the learning rate is set to $2 \times 10^{-5}$.

**Metrics**  We follow prior work (Xu et al., 2024) to have the following metrics: ➊ Effectiveness: whether the model can output the fingerprint target $y$ given the fingerprint trigger $x$. ➋ Reliability: given a random input, the model should not output the fingerprint target $y$ to minimize the false positives. ➌ Efficiency: the training overhead should be minimal. ➍ Harmlessness: the model performance on standard benchmarks should not be degraded. ➎ Persistence: the fingerprint should be persistent after fine-tuning on other datasets.

**Baseline Methods**  We compare our methods with the following baselines: 1) IF$_{\text{SFT}}$: Supervise fine-tune the model based on the default implementation in Xu et al. (2024). 2) UTF$_{\text{IF}}$: Use the fingerprint pair generated by randomly selecting Chinese, Japanese characters and arbitrary model vocabulary tokens as mentioned in Xu et al. (2024). 3) UTF$_{\text{random}}$: Randomly select tokens from the model vocabulary for our method. 4) UTF$_{\text{dialogue}}$: Inspired by Xu et al. (2024), we add readable chat dialogue to both $x$ and $y$.

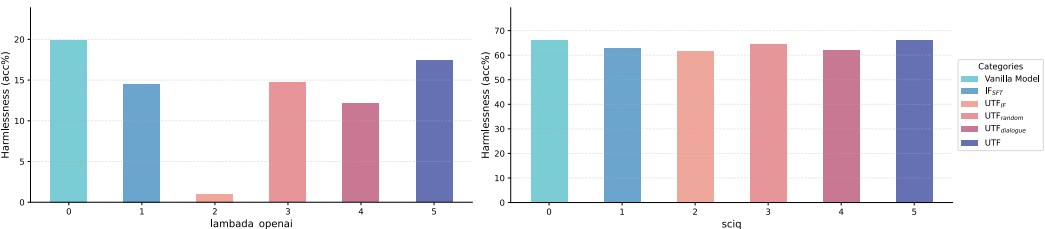

Figure 3: **Harmlessness of UTF and baseline methods on two benchmarks.** The values in this figure represent the test accuracy on LAMBADA OpenAI and SciQ dataset.

## 3.2 RESULTS

**Effectiveness** We evaluate the effectiveness of our methods and baseline methods by inspecting whether the model can output the fingerprint target $y$ given the fingerprint trigger $x$, and the results are shown in Table 1. From the table, we can see that both UTF and baseline methods can have perfect effectiveness on most of the models. This indicates that full-parameter fine-tuning can be an effective method to embed fingerprints into the model. However, we also notice that UTF $_{random}$ cannot embed the fingerprint into Vicuna-7B and Gemma-7B, potentially the random selection of tokens makes it challenging for the model to learn the mapping between $x$ and $y$, especially for some well-established tokens in the model vocabulary.

**Reliability** The reliability is measured by giving 500 random inputs to the model, and the ratio of the model not outputting the fingerprint target $y$ is reported in Table 1. For methods that has no effectiveness on fingerprinting, we use '-' to represent since it cannot generate $y$ even given $x$. From the table, we can see that UTF is the most reliable method, which only has 94.4% reliability on Llama-2-7B-chat and 100% on AmberChat and Gemma-7B. This means that the model will not output $y$ accidentally for most of the inputs.

**Efficiency** The efficiency is measured by the time cost of embedding the fingerprint into the model, and the results are shown in Table 1. We can see that UTF and its variants are the most efficient, which only costs around 6-26 minutes to embed the fingerprint into the model. This indicates that UTF is a highly efficient method for fingerprinting.

**Harmlessness** We evaluate the harmlessness of our methods on two benchmarks: LAMBADA OpenAI (Paperno et al., 2016) and SciQ (Welbl et al., 2017) with zero-shot setting. The results are shown in Figure 3. We can see that our methods have minimal impact on the model performance compared with the vanilla model.

**Persistence** The datasets we use for downstream fine-tuning are: GSM8K (Cobbe et al., 2021), Dolly 15k (Conover et al., 2023), ShareGPT 100k (ShareGPT, 2023), and Aya 200k (Singh et al., 2024). These datasets cover a wide range of scenarios, including math problems and multilingual dialogues. As shown in Table 2, after fine-tuning Llama-2-7B-chat on these datasets, the model can still remember the fingerprint and output the fingerprint target $y$ given the fingerprint trigger $x$. This indicates that the fingerprint is highly persistent after fine-tuning on large datasets, due to the nature of under-trained tokens that are rarely used for the fine-tuning process. More details are discussed in §B.0.1.

| | GSM8K | Dolly 15k | ShareGPT 100k | Aya 200k |
|---|---|---|---|---|
| Llama2-7B-Chat | 100% | 100% | 100% | 100% |

Table 2: **Persistence for Llama2-7B-Chat, after fine-tuning on 4 different datasets.** Values are the Fingerprint Successful Rate (FSR) after we fine-tune fingerprinted models on corresponding datasets.

## 4 LIMITATIONS AND DISCUSSION

There are some limitations to our work. First, due to the computation resource limitation, we do not do large-scale experiments to evaluate other larger LLMs, such as Llama-3-70B (AI@Meta, 2024) and Mixtral-8x7B (Jiang et al., 2024). Second, the malicious user could infer the usage of UTF after seeing the discovery of this work, and it would make it easier to brutally search for the fingerprint input $x$.

We believe that our findings could go beyond the scope of full-parameter fine-tuning. For example, we could adapt the usage of under-trained tokens for adapter-based fingerprinting methods (Xu et al., 2024) to make it more reliable. We leave this as an open question for future research.

## 5 CONCLUSION

In this work, we propose a novel method for fingerprinting large language models using under-trained tokens. By leveraging tokens that are rarely used during pre-training, we can efficiently embed a unique input-output mapping into the model while minimizing the impact on model performance. Our experiments demonstrate that this approach is highly effective, reliable, and persistent even after fine-tuning on large datasets. Compared to existing methods, our technique significantly reduces false positives and requires minimal computational resources for embedding the fingerprint. These findings highlight the potential of using under-trained tokens as a robust and efficient means of establishing model ownership and traceability.

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

## A LICENSE

In this work, we have utilized publicly available datasets and code that are released under specific licenses. We ensure compliance with these licenses and provide appropriate citations for the use of their data and code. For the code we have created, we release it under the MIT license to facilitate broad use and distribution within the research community.

## B ADDITIONAL EXPERIMENTAL RESULTS

### B.0.1 PERSISTENCE AGAINST FINE-TUNING

We fine-tune our fingerprinted models on 4 datasets: GSM8K (Cobbe et al., 2021), Dolly 15k (Conover et al., 2023), ShareGPT 100k (ShareGPT, 2023), and Aya 200k (Singh et al., 2024). For Llama-2-7B-Chat, Vicuna-7B-v1.5 and AmberChat, the fingerprint mapping remains resilient after fine-tuning. For GSM8K and Dolly, we train 3 epochs with learning rate $2 \times 10^{-5}$. For ShareGPT and Aya, we train 1 epoch with learning rate $2 \times 10^{-5}$.

