# OpenReview forum: "UTF: Undertrained Tokens as Fingerprints —— A Novel Approach to LLM Identification"
_ICLR.cc/2025/Workshop/BuildingTrust — Submitted to BuildingTrust_

### Official Review · Reviewer_YvbM · 2025-02-16
**Fingerprinting paper, a little incomplete**

**Rating:** 4
**Confidence:** 4

**Review:**

# Summary

- This paper discusses the field of fingerprinting, which is used to verify model ownership.
- The proposed method, UTF, involves training to associate pairs of tokens (x, y) that are not sufficiently trained during pre-training. During inference, when x is the input, it becomes easier to output y.

# Strengths

- Compared to existing research by Xu et al., this paper achieves improvements in many aspects, including black-box access, minimal impact on performance, reduced training time, robustness to subsequent additional training, and reduction of false positives.
- The evaluation is comprehensive, covering five aspects: Effectiveness, Reliability, Efficiency, Harmlessness, and Persistence.

# Weaknesses and Unclear Points

- Structure: With only a little over five pages out of the allowed nine, the paper feels incomplete. Additionally, there are several structural issues:
  - On Page 3, Figure 2 occupies too much space.
  - It is difficult to find the text for Section 2.1 around line 135 on Page 3.
  - The numbers (0-5) in the labels of Figure 3 are meaningless.
  - Around line 199 on Page 4, the explanation of Baseline Methods is insufficient, and the settings for each are unclear. It is not clear what "IF" stands for.
- Organization of Related Literature: Throughout the text, it sounds like mentioning multiple prior studies, but only a comparison with Xu et al. is made. A comprehensive review of the fingerprinting field is necessary. Additionally, it would be beneficial to clearly distinguish this field from adjacent fields such as watermarking, and discuss why fingerprinting is important.


Overall, while the paper addresses significant issues regarding model ownership, it fails to effectively convey the importance and merits of its methods, and thus, I determined it does not meet the acceptance threshold. By fully utilizing the allowed page count to enhance explanations of related research and methodologies, this could become a good paper.

---

### Official Review · Reviewer_invm · 2025-02-24

**Rating:** 7
**Confidence:** 5

**Review:**

## Summary
The paper proposes using under-trained tokens to construct fingerprint strings for LLMs. This leverages ideas from prior work which finds undertrained tokens by inspecting the unembedding matrix. The under-trained tokens are concatenated to create a fingerprint string, which the LLM is then trained with. The paper demonstrates that such fingerprints are faster to embed, more harmless and persistent after fine-tuning.

## Strengths
* The method is very intuitive, and a straight-forward extension of works in both fingerprinting and LLM identification (i.e. under-trained tokens)
* The empirical results are promising.
* The paper is well-written overall

## Weaknesses
* I find it a bit hard to believe that the effectiveness of fingerprinting methods is less than 100%. In theory one can train a model long enough to   make it memorize any piece of text. Why is it that certain fingerprints are not reliably memorized then?
* The baselines are not clearly described in the paper. Specifically, I could not understand the difference between IF and UTF_{IF}.
* An analysis of $\tau$, which is the hyper-parameter constrolling what fraction of under-trained tokens are used will be insightful.
* An important security risk not considered by the paper is input-filtering, where the input might be blocked because it contains gibberish text. This is a practical attack which is outside the scope of the paper.

---

### Official Review · Reviewer_F3jw · 2025-03-01
**The proposed fingerprinting method shows promise with its straightforward approach, but lacks of important details,**

**Rating:** 5
**Confidence:** 3

**Review:**

The approach leverages tokens that appear infrequently in training data. The proposed method creates a fingerprinting mechanism that exploits the model’s limited exposure to these tokens during pre-training. The authors aim to address fundamental limitations in existing fingerprinting techniques: non-realistic scenario of white-box access during validation, utility trade-off, or effectively embedding the fingerprint.

The paper clearly presents its methodology, making the technical approach accessible and potentially reproducible. The straightforward nature of the technique enhances its appeal for practical deployment scenarios.

Weaknesses:
- While four models were tested, Table 3 only reports comprehensive results for Llama2, raising questions about the method’s generalizability across model architectures.
The experimental protocol lacks details. For instance, the implementation details are ambiguous, particularly concerning the frequency and pattern of how many times the fingerprint is repeated. Also, how does this hyperparameter impact the method?
Exploring other adaptation techniques, such as LoRA, would enhance the work’s practical utility in real-world deployment scenarios.

---

### Decision · Program_Chairs · 2025-03-04

**Decision:**

Reject

**Comment:**

The paper lacks comprehensive experimental validation, as most results focus on Llama2, raising concerns about generalizability across different model architectures. Additionally, key implementation details, such as the frequency and impact of fingerprint repetition, are ambiguous, and the paper does not sufficiently clarify baseline methods, structural organization, or security risks like input filtering.